# Territorialization of Public Action and Mountain Pastoral Areas—Case Study of the Territorial Pastoral Plans of the Rhône-Alpes Region, France

Carine Pachoud 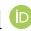

Department of Geography, Innsbruck University, Innrain 52f, 6020 Innsbruck, Austria;
carine.pachoud@univ-grenoble-alpes.fr

**Abstract:** Territorialization aims at improving the effectiveness of public action by adapting to local contexts and including a wide diversity of actors. In the 2000s, the French local authorities, with the support of the European Agricultural Fund for Rural Development (EAFRD), launched more transversal and bottom-up policies on the development of mountain pastoral territories in order to counter national and European sectoral and top-down policies. This article focuses on the Territorial Pastoral Plans (TPPs), a policy of the Rhône-Alpes region, which funds projects defined collaboratively between multiple actors in pastoral territories. The objective is to shed the light on the implementation modalities of the TPPs, and to understand the strengths and weaknesses of this policy in terms of governance to respond to the sustainability challenges of the Rhône-Alpine pastoral territories. A document analysis was achieved and interviews were conducted with nine key actors from four pastoral territories. Results showed that awareness-raising and mediation projects are becoming increasingly important because of the growing conflicts linked to the multi-purpose use of these lands and to wolf predation. Moreover, the integration of environmental actors allows better consideration of ecology in projects. However, the current budgetary restrictions limit their capacity of action within the policy.

**Keywords:** governance; public action; pastoralism; mountain; Rhône-Alpes region

## 1. Introduction

In the paradigm of production-oriented agriculture, which leaves little room for territorial differentiation, mountain pastoralism appears as "a marginal activity in marginal territories" [1] (p. 35). However, faced with the challenge of ecological and social transformation of agriculture, pastoralism has managed to maintain a close link with the territory and aspires to sustainable nature–society relations. This activity is characterized by the seasonal valorization of the mountain grassland resource, most often associated with collective management and the public nature of the property [2]. Pastoralism provides multiple services to society, such as maintenance of landscapes and the social fabric, management of natural risks or even tourism attractiveness [3]. In addition, livestock grazing is often linked to development of quality food value-chains, especially cheese production [4]. This activity ensures multiple economic, social and environmental functions [5]. The quality and distinctiveness of the cheese, most of the time made from raw milk, confer an added value upon milk and cheese often becomes an identity and cultural object [6]. Specific policies are therefore instrumental to support sustainability challenges of pastoralism, generally associated with high-quality food production [7].

In 1962, the implementation of the Common Agricultural Policy (CAP) in the European Union was based on competitiveness objectives through the modernization and industrialization of agriculture [8]. None of the policies provided support to mountain farms which were considered less profitable as compared to lowland farms. The abandonment of mountain farming seemed inevitable [9]. However, its decline has led to

significant environmental and social problems, such as an increase in wastelands leading to landscape modifications and a rise in natural hazards, especially avalanches, as well as massive rural exodus [10,11]. Consequently, this has led to thinking about taking into account the specificities of mountain environments in public policies in order to guarantee the general interest functions provided by pastoralism [2]. Thus, in 1972, the mountain areas in France became the first laboratory for the territorialization of public action through the promulgation of the pastoral law at the national level, and the European decree creating the special mountain allowance. Later in 1975, this became the Compensatory Allowance for Natural Handicaps (CANH). In a context of making agriculture more competitive, these two policies have shown their effectiveness in maintaining mountain livestock farming and are still today the pillars of support measures for pastoralism.

However, these policies remain mainly sectoral and do not allow responding to the new challenges of pastoral territories (environmental issues, multi-purpose use, etc.). Hence, in the 2000s, initiatives with more bottom-up and transversal reflections on mountain territorial development were launched by local authorities. This is the case of the pastoral measures of the second pillar of the CAP. Local authorities, particularly the regions that have become the managing authority of the European Agricultural Fund for Rural Development (EAFRD), have mobilized resources in the co-financing of European policies. In this article, we focus on the policy of Territorial Pastoral Plans (TPP), a Rhône-Alpes regional policy of 2006. This policy aims at supporting a wide variety of projects at the scale of the pastoral territory, defined in a concerted and transversal manner with a diversity of local actors [12]. Thus, this policy appears innovative in terms of the territorialization of public action in favor of pastoralism, by articulating sectoral and territorial policies through bottom-up forms of governance.

The objective of this article is to shed the light on the modalities of implementing TPPs, and to understand the strengths and weaknesses of this policy in terms of governance to meet the sustainability challenges of the Rhône-Alpine pastoral territories. To achieve this objective, an analysis of documents was carried out and semi-structured interviews were conducted with nine key actors from four pastoral territories in the departments of Isère and Savoy. To our knowledge, work on the territorialization of public action in favor of mountain pastoralism is scarce. This is the case in the Pyrenees, with the mountain economy support plan developed in 2006 in the context of the reintroduction of bears [1]. In the Alps, a study was carried out in 2015 on the history of the TPP policy, resulting from the seminar of the French Association of Pastoralism [12]. Our research complements this work, by focusing more on the governance of this policy and its strengths and weaknesses in responding to the current challenges of the Rhône-Alpine pastoral territories.

In the second section, a conceptual framework of the territorialization of public action and examples applied to pastoral territories are exposed. In the third section, the methods and the studied pastoral territories are presented. In the fourth section, the results are outlined, first with a description of the TPPs implementation and then with a more in-depth governance analysis from four pastoral territories. Finally, the discussion and the conclusion are developed in the fifth and sixth sections, respectively.

## 2. The Territorialization of Public Action in Mountain Pastoral Areas

Today, the classic model where a government defines sectoral public policies appears outdated. The aspiration of local populations to govern their territory is growing and, as a result, the question of the relevant scale of decision-making and intervention of public policies becomes a central element. In addition, mobilized actors are multiplying and progressively acting in a transversal manner. Thus, to take into account all of these social and spatial interactions, the term "public action" is now preferred over "public policy" [13].

Regarding the concept of territorialization, there is an epistemological difference between the French-speaking and English-speaking geography. In English-speaking geography, territorialization is often linked to the establishment of state authority over people and resources within these boundaries [14]. However, today this Eurocentric and reduc-

tionist definition of territory, associated with the modern state, tends to be increasingly questioned [15]. In this article, the French understanding of territorialization and the underlying concept of territory are preferred. Territory is understood as a social construct, and not as a politico-administrative area, which is the result of the coordination of actors around shared problems [16]. From this definition, territorialization therefore aims at better considering local contexts to improve the effectiveness of public action. [17] identified three main forms of territorialization of public action:

- Normative territorialization, where the differentiation of public action takes place in a top-down logic, through zoning, to support marginalized areas.
- Territorialization with the support of public action by local authorities (decentralization).
- Pragmatic territorialization, which is based on consultation between diversified actors and the definition of relevant territories delimited according to the problems to be solved. Thus, pragmatic territorialization introduces two elements which are the scale of action (implemented as close as possible to the territory) and transversality (including a great diversity of actors) [18].

In order to take into account the specificities of mountain agriculture and guarantee the functions of general interest provided by pastoralism, two main policies were implemented since 1972 in France, which are the pastoral law and the CANH. These two policies reflect essential differences in the forms of territorialization of public action [2]. First, the pastoral law, defined at the national level, recognizes the specificity of traditional modes of collective management of mountain pastures. It offers tools to local actors, such as Pastoral Land Associations (PLA—land owner groups) and Pastoral Groups (PG—breeder collectives) [19]. In this sense, the pastoral law can be considered as a form of pragmatic territorialization [8]. Second, the CANH, a European scheme, aims at supporting the income of mountain farmers who are not competitive with lowland farmers, in order to remunerate them for services provided to the society. It corresponds to direct aid paid to breeders in areas characterized by natural handicaps (slope, altitude, climate, etc.), according to a rather top-down compensatory and redistributive logic. The CANH is therefore a form of normative territorialization where the consideration of the specificities of mountain territories is the result of decisions taken at the national and European level. Today, the CANH is the main policy for financially supporting French mountain pastoralism [8].

Regarding the TPPs, they seem to offer important windows of action for the implementation of pragmatic territorialization through bottom-up and transversal governance structures. The following section presents the methods of data collection before further analyzing this policy in detail in the results and discussion sections.

## 3. Methods and Case Study

### 3.1. Methods

Information relevant to the functioning and characteristics of the TPPs was first collected by an analysis of documents from action programs, territorial diagnostics or synthesis produced by the territorial institutions coordinating TPPs (Regional Nature Park (RNP), local authorities, etc.) or pastoral services. The former are departmental associations bringing together mountain pasture managers, land owners and local elected officials, created under the pastoral law. Pastoral facilitators play a role of auxiliary to collective pastoral structures, through support for the management of pastoral areas, the implementation of pastoral policies and the provision of job offers and training.

To complete the information, semi-structured interviews were conducted with nine key actors during a field session led during the summer of 2020. The interviewed actors are part of different structures involved in the TPP steering committees of four pastoral territories selected for analysis.

Interviews were first conducted with the director and a facilitator of the Pastoral Services of Isère (PSI), as well as with a facilitator of the Pastoral Services of Savoy (PSS). Their consultation allowed in particular to understand the emergence, evolution and functioning of TPPs, the main characteristics of the TPP of each of the two departments, the

governance system (including the level of participation, trust and conflict), strengths and weaknesses of this policy, as well as forecasting future needs. Moreover, these first three interviews made it possible to select four other TPPs in the two departments, presenting interesting differences for further analysis.

For each selected TPP, more in-depth interviews were carried out with a person from the institution coordinating the TPP. The interviewed persons included a facilitator of the Chartreuse Regional Nature Park (CRNP), a facilitator of the Community of Municipalities of the Vercors Massif (CMVM), a facilitator of the Bauges Regional Nature Park (BRNP) and the director of the assembly of the Community of Municipalities of Tarentaise Vanoise (CMTV).

Finally, interviews were carried out with environmental associations participating in the steering committees of the Isère TPPs, including an engineer from the Isère Natural Spaces Conservatory (INSC) and an another one from the Departmental Federation of Hunters of Isère (DFHI). These two persons were identified, during previous interviews, as relevant partners of the steering committees and the associations they represent. These structures express opinions on environmental projects and are mainly involved in landscape maintenance, biodiversity preservation and infrastructure construction projects.

To organize the meetings, emails were sent to each of these persons to schedule the interview. Each interview lasted for an average of two hours. Table 1 presents the content of the interviews with each actor.

**Table 1.** Content of the interviews conducted with the different actors.

| Institution | Interviewed Actors | Content |
|---|---|---|
| Pastoral Services of Isère | Pastoral services | 1 director 1 territorial facilitator | - history of the institution<br>- current structure and staff<br>- conducted activities and projects<br>- characteristics of the pastures in the department (area, animals, PG, PLA, number of breeders and origin, number of employed shepherds, property regime)<br>- history of the TPP, evolution of the governance<br>- governance structure of each TPP (composition of the steering committee, project holders, facilitator, type of projects, level of participation, trust, conflicts)<br>- funding |
| Pastoral Services of Savoy | | 1 territorial facilitator | - strengths and limits<br>- future needs |
| Chartreuse Regional Nature Park | TPP coordinator | 1 territorial facilitator | - history of the institution<br>- current structure and staff<br>- conducted activities and projects<br>- characteristics of the pastures in the territory (area, animals, PG, PLA, number of breeders and origin, number of employed shepherds, property regime)<br>- history of the TPP<br>- governance structure of each TPP (composition of the steering committee, project holders, facilitator, type of projects, level of participation, trust, conflicts)<br>- funding<br>- strengths and limits<br>- future needs |
| Community of Municipalities of the Vercors Massif | | 1 project manager | |
| Assembly of the Community of Municipalities of Tarentaise Vanoise | | 1 director | |
| Bauges Regional Nature Park | | 1 territorial facilitator | |
| Isère Natural Spaces Conservatory | Environmental association | 1 territorial facilitator | - history of the institution<br>- current structure and staff<br>- conducted activities and projects<br>- role in the TPP and participation level, expertise, project holders<br>- governance dynamics within each TPP (participation, trust, conflicts, other characteristics)<br>- strengths and limits<br>- future needs |
| Departmental Federation of Hunters of Isère | Hunter federation | 1 project engineer | |

### 3.2. Presentation of the Studied Pastoral Territories

For this study, four pastoral territories located in the departments of Isère and Savoy were selected. These territories are the Chartreuse massif, Vercors Quatre Montagnes, the Bauges massif and Tarentaise Vanoise (Figure 1).

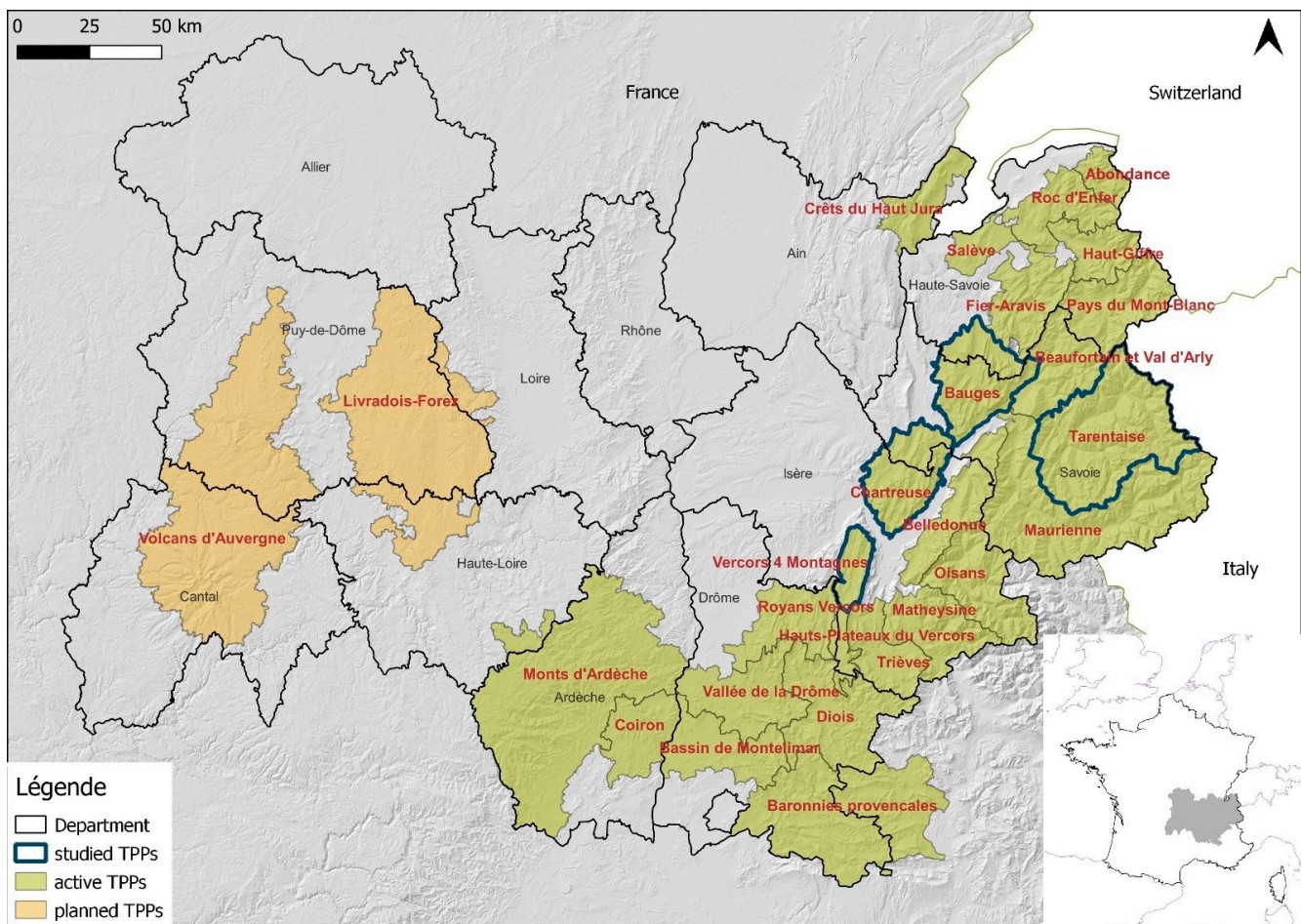

**Figure 1.** Location of the territorial pastoral plans in the Auvergne-Rhône-Alpes region (source: own elaboration).

The two departments show interesting differences in terms of pastoral dynamics (Table 2). First, Savoy has about twice as many pastoral areas and animals moved seasonally to alpine pastures (i.e., the summer pastoral areas located above the permanent residential area) as Isère. The Isère alpine pastures are mainly public and under collective management, while the Savoy mountain pastures are more private and under individual management yet with differences between the territories. In both departments, the mid-mountain pastures (i.e., the pastoral areas used in spring and autumn and located at the level of permanent dwellings) are mostly private. In order to overcome the problem of individual management and private property in Savoy for obtaining financing, many breeders have created mountain pasture Collective Agricultural Interest Companies (CAICs). In Isère, only one CAIC was created in Chartreuse. Its purpose is the development of projects on mid-mountain pastures.

In Isère, two-thirds of the herds are sheep from other departments (mainly from the south of France), and the remaining third consists of mainly beef cattle from the department. Savoy produces many types of cheese with a protected designation of origin (PDO). More than half of the summer herds are dairy herds and 50% of them are lactating cows. The rest are sheep from other departments, as well as some dairy goats from local farms. In addition, there are around two hundred cheese processing units on the Savoy mountain pastures, while there is only one in Isère, located in the Chartreuse massif. Cheese production is labor-intensive and requires more employed shepherds.

**Table 2.** Characteristics of the studied departments and territories [20].

| | | Department | | Pastoral Territory | | | |
|---|---|---|---|---|---|---|---|
| | | Isère | Savoy | Chartreuse Massif | Vercors Quatre Montagnes | Bauges Massif | Tarentaise Vanoise |
| Total area (ha) | | 786,423 | 626,274 | 39,637 | 25,523 | 87,901 | 177,442 |
| Alpine pasture area (ha) | | 66,855 | 134,661 | 3680 | 3618 | 6500 | 57,020 |
| Stocking density (LU/ha) | | 0.39 | 0.42 | 0.39 | 0.39 | 0.44 | 0.37 |
| Percentage of alpine pastures under collective management | | 70.5 | 34.3 | 70.6 | 91.1 | 17.2 | 48.3 |
| Percentage of alpine pastures under public ownership | | 68.6 | 47.8 | 60.0 | 100 | 32.5 | 80.0 |
| Mid-mountain pastures area (ha) | | 11,876 | 21,002 | 4000 | 592 | 5500 | 6988 |
| Percentage of mid-mountain pastures under public ownership | | 6.1 | 9.8 | 0 | 4.9 | 7.3 | 13.5 |
| Number of PG | | 88 | 80 | 10 | 9 | 7 | 43 |
| Number of PLA | | 25 | 40 | 3 | 0 | 4 | 7 |
| Type of animals (%) | Sheep | 67 | 42 | 39 | 65 | 3 | 39 |
| | Dairy cattle | 1 | 29 | 5 | 0 | 30 | 37 |
| | Other cattle | 31 | 27 | 56 | 35 | 61 | 22 |
| | Goat | 1 | 2 | 0 | 0 | 6 | 2 |
| Number of employed shepherds | | 125 | 250 | 7 | 6 | 0 | 160 |

Adapted from Ref. [20].

## 4. Results

In the first subsection, a general description, including the history, the functioning and the funding of the TTP policy is exposed. Then, in the second subsection, a more in-depth governance analysis from the four studied pastoral territories is carried out.

### 4.1. The Territorial Pastoral Plans: A Rhône-Alpes Policy for the Sustainable Development of Pastoral Territories

In this section, a description of the TPP policy (i.e., history, implementation, funding), obtained from the document analysis and the interviews of pastoral service actors, is exposed in order to understand how it is implemented.

#### 4.1.1. History

Since 1972, the Rhône-Alpes region, and to a lesser extent its departments, have secured funds for pastoral improvements carried out by collective structures (municipalities, PG, PLA, etc.). However, this policy remained very sectoral because it did not integrate all the actors in the territory but rather focused mainly on pastoral facilities [12]. Since the 1990s, in a context of growing considerations of environmental issues, problems linked to the growing predation of wolves and tourism pressure, as well as the increase in European budgets for pastoral areas, the region was called upon to develop a more territorial and transversal approach. Consequently, the pastoral policy No. 06.05.883 of 2006 related to the development of pastoral territories, under the leadership of Jean-Jack Queyranne, anticipated the establishment of TPPs. This policy aimed at making pastoral policy more open to all actors in pastoral areas (breeders, local elected officials, foresters, tourism actors, environmental associations, etc.) in order to allow multi-purpose use development of these areas and to better meet local needs.

Since the creation of the policy, twenty-five TPPs have emerged in the region. The new region is planning to create two TPPs in Auvergne (Figure 1).

#### 4.1.2. Functioning

The pastoral territory is defined by the initiative of local actors, most often at the scale of a massif, a valley or an RNP. It can be located between two departments. For each territory, there is an institution coordinating the TPP, leading and organizing the consultation. Pastoral services aim at supporting project leaders on a technical level and can coordinate with the leading institution through the provision of services. TPP contracts are signed on the basis of a prior territorial diagnosis, often entrusted to pastoral services, in order to identify local issues.

A steering committee brings together the various actors in the territory to assess the projects that are carried out. This committee has an advisory role to the region and to the EAFRD. It is also equally divided among a group of breeders, elected officials (municipalities and communities of municipalities) and partners (RNP, environmental associations, tourism and cultural actors, foresters). Representatives of the region and departments are also present as funders. The steering committees take place at least once a year. Prior to their meetings, summary sheets for each project are shared with the members of the steering committee to inform them about the projects and ask them to add any comments or recommendations. Project leaders are only collective actors. These are mainly the municipalities, the PLAs, the PGs and the CAICs, as well as to a lesser extent the RNPs, pastoral services and environmental associations. Technical committees of experts (pastoral services, RNP, etc.) can take place prior to the steering committees in order to further discuss technical projects.

Since 2017, a selection committee with deliberative power has been created. This committee gathers technicians representing the region and the EAFRD. Based on the advisory opinion of the steering committee, this selection committee meets several times a year to rate the projects according to a selection grid.

#### 4.1.3. Funding

The objective of the TPPs is to fund all the needs related to collective pastoralism within the region. It is therefore a matter of determining for each project, at the territorial level, the share of self-financing from local structures holding projects and the European, regional and departmental co-financing. The region, which is the managing authority of the EAFRD, is widely mobilized in the co-financing of European policy with regard to pastoral measures (measure 7.61, which supports investments allowing the maintenance and enhancement of pastoral areas, and measure 16.71, which supports the implementation of local development strategies). Over a period of five years, the region commits to funding 70% of pastoral projects (35% region and 35% EAFRD); 80% of studies, awareness-raising and mediation actions (40% region and 40% EAFRD); and 100% of the creation of collective structures (PG, PLA). In addition, departments have specific lines of funding which vary among departments. Thus, the department of Isère finances up to 75% of PG and PLA projects, and replaces the regional and EAFRD funds. In total, the department subsidizes 24.6% of the projects, while the department of Savoy finances an additional 10% to the region and the EAFRD for projects led by the PLA. This represents 2.7% of the projects (Table 3).

**Table 3.** Contributions in euros from the various funders of projects defined under the TPPs for the departments of Isère and Savoy in 2019 (source: PSI, personal communication).

| Department | Region | EAFRD | Department | Self-Financing | Total |
|---|---|---|---|---|---|
| Isère | 217,018 (23.8%) | 217,018 (23.8%) | 224,161 (24.6%) | 254,312 (27.9%) | 912,509 |
| Savoy | 497,492 (28.2%) | 544,620 (30.9%) | 47,128 (2.7%) | 673,439 (38.2%) | 1,762,678 |

In total, an average of 80% of funding is intended for pastoral improvement (infrastructures, bush clearing, etc.), 10% for transversal issues (diagnostics, studies and

communication) and 8% for facilitation and coordination. While municipalities generally have needs for heavier equipment such as for the construction of chalets or tracks, PGs and PLAs carry less expensive work such as fencing, bush clearing and animal watering troughs. The partners, and elected officials, to a lesser extent, are more involved in transversal projects.

In the following section, a more in-depth analysis of governance processes will be carried out in four pastoral territories where the TTP policy is implemented.

*4.2. In-Depth Analysis from Four Pastoral Territories*

The results from the different interviews led in four pastoral territories, the Chartreuse massif, the Vercors Quatre Montagnes, the Bauges Massif and the Tarentaise Vanoise, are presented consecutively. From these interviews, an analysis of the strengths and weaknesses of the TPP policy in terms of governance is developed consequently.

4.2.1. Divergent Governance Dynamics

In the Chartreuse massif, located between Isère and Savoy, the TPP is coordinated by the RNP, which has taken a strong interest in the pastoral issue. However, the park must face cuts in public funding which weaken its role of facilitating and supporting projects. In this massif, the majority of projects concern pastoral improvements and are located in Isère. The territory part located in the Savoy department rarely develops projects due to the presence of only one PG. The PSI is therefore the privileged pastoral service and is involved in supporting projects that require technical expertise. The awareness-raising projects are led by the park and cover the entire territory. Most of the breeders are local and know each other well. However, the results show that the participation in the steering committees is low due to the participation of the projects holding actors only. The actors of the park regret the lack of associative actors, especially environmental actors on board of the steering committees. According to the territorial facilitator, they are facing budgetary restrictions that limit their interventions. As a result, the environmental assessment of projects is mostly carried out by the park itself.

The Vercors Quatre Montagnes TPP, located in Isère, is coordinated by the CMVM and is co-facilitated by the PSI. The interviewed actors reported that the steering committees are dynamic and that there is a good diversity of participating actors (breeders, elected officials, partners). The majority of cattle farmers are local and know each other well, which facilitates their participation and communication. However, sheep farmers from other departments are not present in the steering committees due to geographical distance. The CMVM often carries out original awareness-raising projects, such as making films. The PDO cheese *Bleu du Vercors* plays an important role in the pastoral economy. Nevertheless, it is mainly heifers that are put on the alpine pastures, while lactating cows remain on the lowland permanent farms.

In the Bauges massif, the TPP is coordinated by the RNP. However, similarly to the CRNP, it has to cope with decreases in funding which leads to a reduction in the time dedicated to the facilitation and coordination of the TPP. The PSS sometimes intervenes for a technical support to the projects. The territory is confronted with a growing abandonment of mountain pastures. The animals moved to alpine pastures are mostly heifers, while lactating cows are staying more and more on the permanent farm. However, the PDO cheese *Tome des Bauges* remains central to the economy of the area. Some breeders carry out milk production on the alpine pastures and others are involved in cheese processing. However, the offer is not segmented and milk from summer pastures does not add value to the cheese. Regarding the steering committees, the interviewees noted a good diversity and participation of actors. The breeders are local and know each other well, and the projects mainly concern pastoral improvements. The RNP is located between the departments of Savoy and Haute-Savoie. However, TPP funds are more mobilized in Savoy because Haute-Savoie offers significant allocations to the national Sensitive Natural Area (SNA) policy for mountain pastures, which replaces the TPP. The SNA policy is a tool for the protection of natural spaces, established in

France since 1976. It is financed by the department through a tax based on urban planning authorizations to compensate for the artificialization of soils. Competition between policies seems to result in a lack of coordination between the two departments.

Finally, the Tarentaise Vanoise territory, located in Savoy, mobilizes an important budget because of its large area. The TPP is coordinated by the CMTV and is co-facilitated by the PSS. The PDO cheese *Beaufort* holds a vital role in the territorial economy. Many mountain pastures carry out milking of dairy cows which requires significant pastoral infrastructures (tracks, chalets, water points, etc.), most often developed by the municipalities. The creation of tracks is often a conflictual subject between breeders, on the one hand, who would like to build more tracks to facilitate access to the pastures, and other actors, especially environmental ones, on the other hand, who wish to limit infrastructures. According to the interviewees, this territory has undergone a large process of artificialization by ski resorts and many actors now wish to limit the human impact on mountains. Moreover, the PSS carries out mediation projects with tourism actors. As per the interviewees, participation in the steering committees is good and with fair representativeness of the diversity of actors. During the summer, a steering committee is organized on the field with the various actors to present achieved or future projects in order to facilitate consultation.

Table 4 presents the main characteristics of the four studied TPPs.

**Table 4.** Characteristics of the four studied TPPs.

| | Chartreuse Massif | Vercors Quatre Montagnes | Bauges Massif | Tarentaise Vanoise |
|---|---|---|---|---|
| Institution in charge of the TPP | CRNP | CMVM | BRNP | CMTV |
| TPP dates | 2010–2015 2017–2021 | 2010–2015 2016–2020 | 2008–2013 2015–2020 | 2010–2015 2017–2021 |
| Average number of projects per year | 7 | 5 | 6 | 15 |
| Types and description of projects | - Improvements (tracks and chalet construction, bush clearing) <br> - Awareness-raising (brochures, educational kits toward tourists) | - Improvements (tracks and chalet construction, bush clearing) <br> - Awareness-raising (films, display panels, shared meals with breeders) | - Improvements (tracks and chalet construction and renovation, construction of animal watering troughs and water catchment basins) | - Improvements (tracks and chalet construction and renovation, bush clearing) <br> - Mediation actions (meetings with tourism actors (bike and ski resorts)) |
| Main project leaders | Municipalities, community of municipalities, CAIC, PG, RNP | Municipalities, community of municipalities, PG, PSI | Municipalities, community of municipalities, CAIC, RNP | Municipalities, CAIC, PLA, CMTV, PSS |
| Number of steering committees per year | 2 | 2 | 2 | 3 |
| Regional funding for the second TPP (in EUR) | 169,000 | 100,000 | 238,000 | 1,000,000 |
| Main governance characteristics | - Cuts in public funding which weaken the scope of action of the RNP and lead to low participation of environmental actors <br> - Majority of projects in Isère <br> - Participation of projects holding actors only in steering committees | - Good participation of each type of actor, but sheep breeders are not present because they are distant <br> - Original awareness-raising actions <br> - PDO cheese *Bleu du Vercors* important for the economy but not valorized through alpine pasture use | - Cuts in public funding which weaken the scope of action of the RNP <br> - Good participation of each type of actors, breeders are local <br> - PDO cheese *Tome des Bauges* important for the economy but not valorized through alpine pasture use <br> - Competition between the TPP policy in Savoy and the SNA policy in Haute-Savoie | - Important infrastructures required for cheese production, which can be a source of conflicts <br> - Mediation projects central due to large tourism activities <br> - Good participation of each type of actors, breeders are local. One steering committee on the field <br> - PDO cheese *Beaufort* central for the economy and valorized through alpine pasture use |

4.2.2. Strengths and Weaknesses of the Policy in Terms of Governance

The TPP policy has made it possible to set up a new form of governance through the integration of the various actors of pastoral territories in the consultation process. Transversality was positively perceived by all the interviewed actors who even expressed their wish to broaden the diversity of actors in the steering committees, particularly by inviting representatives of agri-food value chains (for example dairy cooperatives), shepherds or more representatives of the tourism sector. Moreover, the analysis revealed that the diversification of the actors allowed establishing a certain social control and reducing the risks of clientelism in obtaining funding. In addition, this system provides an environmental assessment of projects and therefore a better consideration of ecology in projects by local actors. However, environmental actors prefer an even greater consideration of environmental aspects starting from the inception phase of the project and through its development, rather than only a consultation after writing the project. The monitoring and evaluation of projects are carried out by the Departmental Directorates of the Territories (DDT), most often accompanied by pastoral services. The environmental actors expressed their willingness to support, in the future, the work on the field in order to avoid deviations from the initial project.

Nonetheless, inclusive governance requires more effort from local actors, especially from the institutions coordinating TPPs and pastoral services. According to a pastoral facilitator, "TPPs are a very nice tool, they made us grow as pastoral services, yet mediation takes a long time". The role of the technician in pastoral services has thus changed toward a more accompanying role. Nowadays, the technician is supposed to follow a bottom-up logic with local actors and intervene in a different way depending on the territorial context.

The interviews showed that the multi-purpose use of pastoral territories is today the main subject of conflict, particularly between shepherds and breeders on the one hand and those who practice sports (hikers, cyclists, etc.) on the other hand. These conflicts also emerge from the reinforcement of devices linked to predation by wolves and the increase in leisure activities in pastoral areas. Pastoral actors expressed users' lack of knowledge on pastoral activity, leading to inappropriate behavior (e.g., forgetting to close fences, passing through the middle of pastures, bad reactions to guard dogs). The interviewed actors expressed their desire, for the future, to increase awareness-raising projects for the users of these territories and to improve mediation with actors in tourism and leisure, such as ski and mountain biking resorts. Some areas have already developed innovative projects, such as the CMVM, which produced an awareness-raising film entitled *Moi le pastoralisme* (I, pastoralism), and the CRNP, which developed educational kits for shepherds to better communicate with hikers.

Finally, the analysis brought to light problems related to funding. First of all, the majority of the actors expressed the lack of means for facilitation and coordination, which impacts the quality of communicating the policy and supporting the projects. In addition, the actors highlighted the problems of budgetary restrictions of RNPs and environmental associations which limit their capacity to intervene within the TPPs. Furthermore, the lack of cash flows in small municipalities appears to be a big limitation as this leads to low efficiency and delays in execution, specifically in pasture rehabilitation. The increase in departmental, regional and European funds therefore appears to be necessary to increase the effectiveness of the policy. Competing with other national or European policies, implemented at a departmental level, has been reported by some actors, as a major issue. Thereby, the SNA policy in Haute-Savoie is highly endowed for alpine pastures while the TPP policy is little used, hence leading to a lack of cooperation with the departments having TPP in common (case of the Bauges massif). Finally, the actors expressed the lack of information on the durability of the TPP policy in the future. For the new CAP of 2021–2027, the region has decided to renew the policy, but the amount allocated has not yet been defined. Most of the actors expressed their fear of seeing regional and European funds decrease, particularly because of budgetary restrictions and the inclusion of Auvergne in

the policy. The region has already communicated the risk of a lack of credits compared to grant requests for the year 2020, which would lead to prioritizing certain projects.

## 5. Discussion

Before the establishment of the TPPs in 2006, the projects implemented in pastoral territories were strictly agricultural ones. The pastoral services worked exclusively with the concerned breeders and municipalities. Today, other actors are appearing around the table, particularly environmental actors. These different actors started working together even though many did not know each other or even had conflicting relationships. The TPPs have thus made it possible to create a space for dialogue and exchange to define common interests [12]. Consequently, this policy has provided the conditions for a pragmatic territorialization of public action by offering local actors an alternative and open form of governance within self-determined pastoral territories.

However, the results revealed that the participation level of the breeders depends mainly on whether they live on the territory or not. The geographical proximity allows them to know each other better and to express themselves more easily during steering committees. Then, it appears that the participation level of the different actors, especially breeders and local authorities, can be improved by original arrangements, such as a yearly steering committee on the field in Tarentaise Vanoise. In addition, the participation level of environmental actors is greatly conditioned by public funding, which today tends to be reduced. After that, coordination of TTPs located on two departments seems more difficult to manage, because pastoral and environmental institutions are different and the support programs for pastoralism can vary and compete.

Furthermore, it appears that the role of local actors could be further strengthened through their integration into the policy review process and through granting them a deliberative role in the choice of projects. In addition, it seems useful to integrate a greater diversity of actors into the policy, including actors from the agri-food value chains and shepherds, in view of their importance in the management of these spaces and communication with tourists. This could also allow better valorization of alpine products [5]. It also appears necessary to further integrate tourism actors given the new challenges linked to multi-purpose use. Indeed, although pastoral improvement projects are still largely a majority, awareness-raising and mediation projects are playing an increasingly important role because of growing conflicts between agricultural actors and new users of the pastoral space. These conflicts are mainly linked to the measures put in place against wolf predation (guard dogs, fences, etc.). Awareness-raising and mediation therefore seem to be crucial points for the future development of pastoral areas, and their financing should play a more important role within the TPP policy. Finally, it also seems desirable to include environmental actors (RNP and environmental associations) more at the level of defining the project as well as at the monitoring and evaluation level. This represents a central point for increasing the environmental sustainability of pastoral areas. However, the neoliberal paradigm which leads to a reduction in public funding for these structures strongly impacts their capacity for action within the policy.

This article aimed at describing the functioning of the TPP policy and at identifying the main governance challenges this policy should meet in order to increase the sustainability of the Rhône-Alpine pastoral territories. However, some limitations of this paper need to be mentioned. For a more in-depth understanding, it would be useful to carry out interviews with other actors of pastoral territories (actors from the tourism and agri-food sectors, breeders, shepherds, etc.) in order to understand their point of view, and also to extend the analysis to other pastoral territories. This work appears even more interesting as the Auvergne, which has a very different context from Rhône-Alpes, integrates the TPP policy. Moreover, it is important to see how this policy risks being impacted by the new CAP.

Finally, it is useful to compare this policy with other pastoralism support policies developed around the world, especially in Europe, in an attempt to improve them. In the European Union (EU), the financial support for pastoralism is mainly administered

through the CAP. The main contributions are related to the CANH and also to specific agri-environmental measures (AEMs), as showed in Austria [21], France [8], Germany [22] and Portugal [23]. Although the framework is common to all the member countries, variations in the application of the different measures are possible [24]. In fact, the principle of subsidiarity allows each member country to establish measures adapted to their context. For example, Bulgaria established a program that supports traditional pastoral breeding systems and promotes the use of local guard dogs [25]. In Greece, some measures encourage tourism activities linked to pastoral farming [26]. There are also some transnational projects, such as the Alpinet Gheet Project, which promotes sheep and goat sector and interaction with tourism at the scale of the Alpine region [27,28]. In addition to the CAP measures, some countries developed their own program for pastoral development. It is the case in Germany, where the Bavarian and the federal governments cover investments to renovate pastoral infrastructures, build routes to alpine pastures or support alpine gastronomy on summer farms [22]. Outside the EU, Switzerland also implemented many original measures to support pastoralism. For example, the environmental contributions integrate the quality of cultural landscapes and biodiversity [29]. Since 2014, the Swiss government also provides a contribution to the breeders in order to encourage them to move their herds to summer pastures [30]. The level of subsidies is calculated according to the stocking density [31]. In addition, Switzerland developed an official designation to protect the origin of products coming from summer pastures [32].

The literature shows that there is a diversity of pastoralism support policies in Europe, alongside traditional CAP measures [24] However, it seems that the TTPs represent an original policy to favor the multi-purpose use of pastoral territories while guaranteeing the maintenance of these fragile areas through a horizontal governance mode. Nonetheless, a deeper analysis of implementation and governance of the different policies is often missing in the literature, which makes comparison difficult. It therefore appears necessary to deepen the comparative analysis through more field work.

## 6. Conclusions

The TPP policy aims at supporting pastoralism in the French Rhône-Alpine region. It enables funding projects in pastoral territories, defined collaboratively between multiple actors (breeders, environmental actors, local authorities, etc.). This policy makes it possible to fund pastoral improvements that local actors could hardly cover (e.g., chalet and track construction). It also allows for improved communication and awareness of the pastoral activity with tourists through original projects. This appears even more relevant given that tourism in pastoral areas is increasing, leading to growing conflicts linked to the multi-purpose use of these lands and to wolf predation. However, pastoral territories present further challenges that need to be consider in the TPP policy. One of them is to integrate shepherds and actors of food value-chains (e.g., dairies), as well as tourism actors, in the governance. Indeed, these actors are important for communication with tourists, and they play a central role for the economy of these territories. In addition, success of the governance in TPPs is linked to different factors, such as the geographical proximity among breeders, which increases their participation. Moreover, the decline in public funding of environmental institutions, including nature parks, limits their scope for action. It therefore seems necessary to increase their budgetary allocations. Finally, it is important to increase coordination among departments sharing a same TPP, because the different pastoral and environmental institutions and possible competition among support programs for pastoralism can decrease the efficiency of the policy.

Nevertheless, even if this policy shows significant potential to fulfil challenges of pastoralism, the implemented agricultural policies still strongly reinforce the top-down logic. Indeed, alternative and territorial policies, such as the TPPs, benefit from limited support [8]. The EAFRD funding for the Auvergne-Rhône-Alpes region represented 2.4 billion euros for the period of 2014–2020, and of which 62% were sourced from CANH. The share for TPP represented less than 0.4% of this budget (around EUR 9 million). These

programs deserve a greater place in agricultural policies because they open the way to more democratic forms of governance and help increase the sustainability of mountain territories, especially the most marginalized.

**Funding:** This research was funded by the HIGHLANDS.3 project (Collective Approach of Research and Innovation for Sustainable Development in Highland) (grant H2020-MSCA-RISE-2019-872328-HIGHLANDS.3).

**Informed Consent Statement:** Informed consent was obtained from all subjects involved in the study.

**Conflicts of Interest:** The author declares no conflict of interest.

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
