# Peer review of "Territorialization of Public Action and Mountain Pastoral Areas—Case Study of the Territorial Pastoral Plans of the Rhône-Alpes Region, France"

_sustainability, doi:10.3390/su13148014_

Round 1
Reviewer 1 Report
Pastoral programs are usually connected with special cheese production in the mountines area. It should be mentioned in this study, as we try to fulfill sustainability challenges in this type of special production in agriculture. It would be more interesting, when results from the survays could be presented on charts.
- 328 - Table 4 should have more detailes concerning: types of provided projects, goals of the projects, results of the projects etc.
- There is no Conclusion at all, so it should be added.
Author Response
Response to Reviewer 1
First of all, I want to thank the Reviewer who helped to improve the quality of the paper.
Point 1: Pastoral programs are usually connected with special cheese production in the mountines area. It should be mentioned in this study, as we try to fulfill sustainability challenges in this type of special production in agriculture. It would be more interesting, when results from the survays could be presented on charts
Response: In introduction from line 34 to 40, I exposed the importance of high-quality cheese production for mountain areas and the need to connect cheese production with pastoral programmes. Then, I presented the results from the surveys in Table 4 to facilitate reading.
Point 2: Table 4 should have more detailes concerning: types of provided projects, goals of the projects, results of the projects etc.
Response: I developed in Table 4 the types and the description of the projects implemented in each TPP.
Point 3: There is no Conclusion at all, so it should be added.
Response: I wrote a conclusion in the sixth section from line 489 to 517.
Reviewer 2 Report
The article entitled “Territorialization of public action and mountain pastoral areas. Case-study of the Territorial Pastoral Plans of the Rhône-Alpes region, France ”deals with an important and very current problem of abandoning agricultural production in marginal (mountain) areas. The author based the article on qualitative analyzes, which is not a common practice. Both the purpose of the manuscript and the methods are appropriate. The results can play a role in strengthening the financing of local CAP projects. The weakest side of the text is the relatively poor literature review. The author does not refer to how such projects are implemented in other parts of the Alps (German, Italian, etc.). In Germany, there is a relatively well-organized agricultural policy supporting mountain agriculture, probably including pastoralism. However, this should be verified by referring to the literature on the subject. The manuscript is well prepared, but please consider the following:
- add a map with the location of the studied area - e.g. against the background of Europe
- complete the information on stocking density in Table 2
- present the limitations of research
- try to improve the literature review.
Author Response
Response to Reviewer 2
First of all, I want to thank the Reviewer who helped to improve the quality of the paper.
Point 1: add a map with the location of the studied area - e.g. against the background of Europe
Response: I made my own map in Figure 1 showing the localisation of the different TPPs, of which the studied TPPs, including a background of Europe.
Point 2: complete the information on stocking density in Table 2
Response: I completed this information
Point 3: present the limitations of research
Responses: I presented limitations of the research in discussion from line 451
Point 4: try to improve the literature review.
Response: in discussion I improved the litterature review from line 459 to 481, in order to make a comparison with other pastoral policies from European countries (especially from Germany, Switzerland, Bulgaria, Greece, Austria, Portugal).
Reviewer 3 Report
Dear author,
thank you for an interesting empirical article. I find that there is not much to improve, however, I believe that a more thorough engagement with the literature on pastoralism could be achieved. The number of references overall is low. How do your findings compare to other local contexts outside France?
Further, I think there is a confusing use of the word "territorialization", which is associated with a different meaning in English-speaking critical geography (e.g. Vandergeest and Peluso 1995). You should provide a clear definition of your use of the term and delineate this towards the common, critical use.
Also, I do not think the word "animation" carries the same meaning in English as in French. You should look for an alternative (engagement? participation? activation?).
The tables are hard to read. Columns should be left-aligned.
Best regards
Author Response
Response to Reviewer 3
First of all, I want to thank the Reviewer who helped to improve the quality of the paper.
Point 1: The number of references overall is low. How do your findings compare to other local contexts outside France?
Response: in discussion I improved the litterature review from line 459 to 481, in order to make a comparison with pastoral policies from other European countries (especially from Germany, Switzerland, Bulgaria, Greece, Austria, Portugal).
Point 2: Further, I think there is a confusing use of the word "territorialization", which is associated with a different meaning in English-speaking critical geography (e.g. Vandergeest and Peluso 1995). You should provide a clear definition of your use of the term and delineate this towards the common, critical use.
Response: in part 2, from line 96 to 106, I exposed the difference between French-speaking and English-speaking geography related to the concept of territorialization. Then, I used a clear definition from French-speaking literature.
Point 3: Also, I do not think the word "animation" carries the same meaning in English as in French. You should look for an alternative (engagement? participation? activation?)
Response : It is true, I used the word "facilitation" instead of animation.
Point 4 : The tables are hard to read. Columns should be left-aligned.
Response: I made the suggestion.